# Restoration of Atmospheric Turbulence-Degraded Short-Exposure Image Based on Convolution Neural Network

Jiuming Cheng [1,2,3], Wenyue Zhu [2,3,*], Jianyu Li [2,3], Gang Xu [2,3], Xiaowei Chen [2,3] and Cao Yao [1,2,3]

1   Key Laboratory of Atmospheric Optics, Anhui Institute of Optics and Fine Mechanics, HFIPS, Chinese Academy of Sciences, Hefei 230031, China; chengjiu@mail.ustc.edu.cn (J.C.)
2   Science Island Branch of Graduate School, University of Science and Technology of China, Hefei 230026, China
3   Advanced Laser Technology Laboratory of Anhui Province, Hefei 230037, China
*   Correspondence: zhuwenyue@aiofm.ac.cn

**Abstract:** Ground-based remote observation systems are vulnerable to atmospheric turbulence, which can lead to image degradation. While some methods can mitigate this turbulence distortion, many have issues such as long processing times and unstable restoration effects. Furthermore, the physics of turbulence is often not fully integrated into the image reconstruction algorithms, making their theoretical foundations weak. In this paper, we propose a method for atmospheric turbulence mitigation using optical flow and convolutional neural networks (CNN). We first employ robust principal component analysis (RPCA) to extract a reference frame from the images. With the help of optical flow and the reference frame, the tilt can be effectively corrected. After correcting the tilt, the turbulence mitigation problem can be simplified as a deblurring problem. Then, we use a trained CNN to remove blur. By utilizing (i) a dataset that conforms to the turbulence physical model to ensure the restoration effect of the CNN and (ii) the efficient parallel computing of the CNN to reduce computation time, we can achieve better results compared to existing methods. Experimental results based on actual observed turbulence images demonstrate the effectiveness of our method. In the future, with further improvements to the algorithm and updates to GPU technology, we expect even better performance.

**Keywords:** atmospheric turbulence; Zernike polynomials; image restoration; convolutional neural networks

## 1. Introduction

The imaging performance of a ground-based remote observation system is degraded due to the influence of atmospheric turbulence. The cause of turbulence is complex and is affected by atmospheric temperature, pressure, humidity, wind speed and other factors [1]. When light waves pass through turbulence, the wavefront phase will be distorted and accumulated. The random distortion of the phase is converted into the corresponding point spread function (PSF). Two driving factors affect turbulent PSF. The first factor is the random tilt that causes the image pixel offset. The other is the higher-order distortion that leads to image blurring [2,3].

The traditional turbulence image restoration method is generally divided into three steps. The first step involves removing the pixel offset caused by tilt. The most commonly used method for this process is the optical flow method, such as Xie et al. [4], Hardie et al. [5] and Gilles et al. [6]. In addition, because the random phase distortion is zero mean value, clear areas occasionally appear in the image [7]. Therefore, the next processing method is to compare the sharpness of the pixel blocks within a certain shooting interval and select the clearest pixel block for fusion. This method is also called lucky image fusion, such as Anantrasirichai et al. [8], Aubailly et al. [9] and Zhu et al. [10]. However, this method requires a large number of measured data to be filtered and fused, and the fusion effect is highly dependent on the quality of measured data. Finally, most literatures use a

blind deconvolution algorithm [8–15] to remove the residual fuzziness after lucky image fusion. However, the PSF priori defined by them in the process of blind deconvolution is too general and not optimized for turbulence characteristics. These priors have little connection with the statistical behavior of turbulence, especially the 5/3 power kernel proved by Fried [16].

Recently, deep learning methods have been gradually applied to turbulence mitigation strategies. Chen et al. [17] proposed a U-net-like deep-stacked autoencoder neural network model. Fazlali et al. [18] proposed an end-to-end convolutional autoencoder to mix several registered blurry frames to generate a high-quality output image of the scene. However, they are usually based on a simplified assumption of atmospheric turbulence where they assume the blur to be spatially invariant. Such an assumption cannot extend to general scene reconstruction. Hoffmire et al. [19] proposed the block matching and CNN (BM-CNN) method. Zhiyuan et al. [20] introduce a physics-inspired turbulence restoration model (TurbNet). Although they performed well in the simulated data set, they did not test and evaluate the measured data.

To tackle the challenges, this paper makes two contributions:

1. We tune a turbulence simulator based on a physical model to generate a large-scale dataset. The highly realistic and diverse dataset provides strong support for the training of our neural network.
2. Realizing the limitations of lucky image fusion and blind deconvolution algorithms, we introduced convolutional neural networks to replace the original two algorithms. Our algorithm reduces the requirements for computation time while ensuring the effectiveness of image restoration.

## 2. Method

### 2.1. Problem Setting and Motivation

The degradation process of atmospheric turbulence on images can be approximated by the following equation:

$$\widetilde{I} = (T \circ B)(I) + n \tag{1}$$

where $\widetilde{I}$ is an image degraded by turbulence, and $I$ is the corresponding clear image. The operation $T$ is a mapping representing the geometric pixel displace (known as the tilt), and $B$ is a convolution matrix representing the spatially varying blur. The operation "$\circ$" means the functional composition. The variable $n$ denotes the additive noise.

Due to the simultaneous existence of the $T$ and $B$ operators in Equation (1), the problem becomes difficult. If there is only $B$, the problem become a deblurring. A common method is to remove the tilt using the reference frame and optical flow method. Equation (1) is simplified as

$$\widetilde{I} = B(I) + n \tag{2}$$

Removing blurring caused by turbulence is an ill-posed inverse problem. From a Bayesian perspective, the solution $I_{MAP}$ can be obtained by solving the Maximum a Posterior (MAP) estimation problem,

$$I_{MAP} = arg\min_{I} -\log p(\widetilde{I}|I) - \log p(I) \tag{3}$$

where $\log p(\widetilde{I}|I)$ represents the log-likelihood of observation $\widetilde{I}$, $\log p(I)$ delivers the clean image $I$ and is independent of degraded image $\widetilde{I}$. More formally, Equation (3) can be reformulated as

$$I_{MAP} = arg\min_{I} \frac{1}{2\sigma^2} \|\widetilde{I} - B(I)\|_2^2 + \lambda R(I) \tag{4}$$

where the solution minimizes an energy function composed of a data term $\frac{1}{2\sigma^2}\|\widetilde{I} - B(I)\|_2^2$ and a regularization or prior term $\lambda R(I)$ with regularization parameter $\lambda$.

Generally, the methods to solve EquationEquation (4) can be divided into two main categories, i.e., model-based and learning-based methods. The former method often requires multiple iterations, resulting in a long calculation time. Therefore, we choose the latter method. The latter mostly trains a truncated unfolding inference through optimization of a loss function on a training set containing $N$ degraded-clean image pairs $\{(\widetilde{I}_i, I_i)\}_i^N$. In particular, the learning-based methods are usually modeled as the following bi-level optimization problem.

$$
\begin{cases}
\min_{\Theta} \sum_{i=1}^{N} L(I_{MAPi}, I_i) \\
s.t. I_{MAPi} = arg\min_{I} \frac{1}{2\sigma^2}\|\widetilde{I} - B(I_i)\|_2^2 + \lambda R(I_i),
\end{cases}
\tag{5}
$$

where $\Theta$ denotes the trainable parameters, $L(I_{MAPi}, I_i)$ measures the loss of estimated clean image $I_{MAPi}$ with respect to ground truth image $I_i$. The learning-based methods employ the defined $I_{MAPi} = f(\widetilde{I}, \Theta)$ to iteratively optimize parameters $\Theta$ through the upper equation of Equation (5)., such that the function $f$ progressively approaches the lower equation of Equation (5).

According to the above description, we use the optical flow method to remove the pixel offset caused by tilt. Afterwards, we remove the blurring effect by convolutional neural networks (CNN). Our pipeline diagram is shown in Figure 1. The detailed algorithm will be presented in the following subsection.

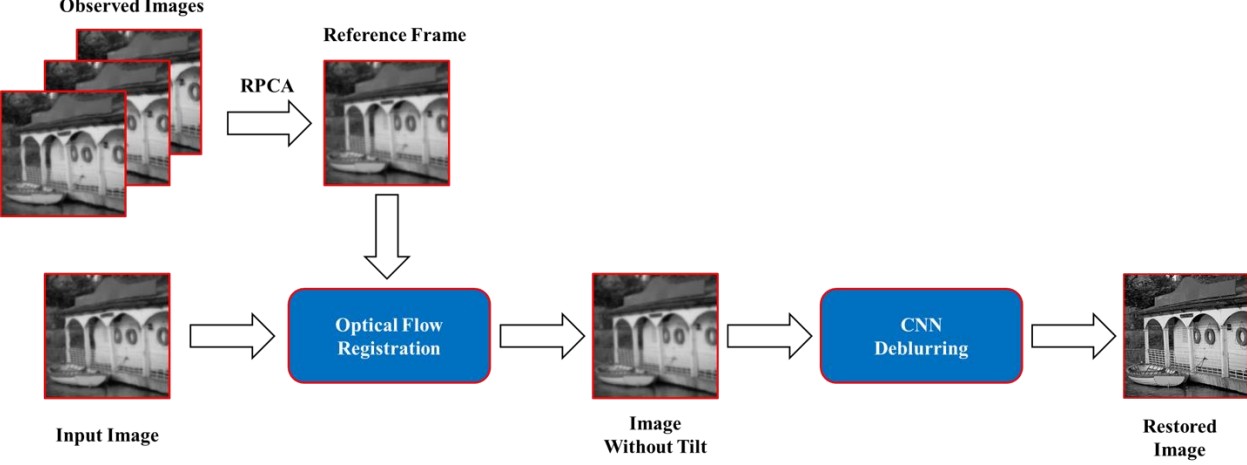

**Figure 1.** Block diagram of the proposed restoration scheme.

### 2.2. Construction of Reference Frame and Optical Flow Registration

Using the optical flow method to register images requires the reference frame. Referring to the method of sparse decomposition for background modeling [21,22], we use robust principal component analysis (RPCA) to perform matrix decomposition to obtain the low-rank component and construct a reference frame for registration. The RPCA low-rank decomposition can be defined as follows:

$$
\text{minimize}\|L\|_* + \lambda\|S\|_1 s.t. L + S = G
\tag{6}
$$

where $L$ is the low-rank component, $S$ is the sparse component, $G$ is the observed image, and $\lambda$ is a constant providing a trade-off between the sparse and low-rank components. A decomposed result is illustrated in Figure 2. Average the low-rank components of multiple images to obtain the final reference frame. After obtaining the reference frame through

RPCA, we referred to the coarse-to-fine optical flow method [23] for registration, as shown in Figure 3.

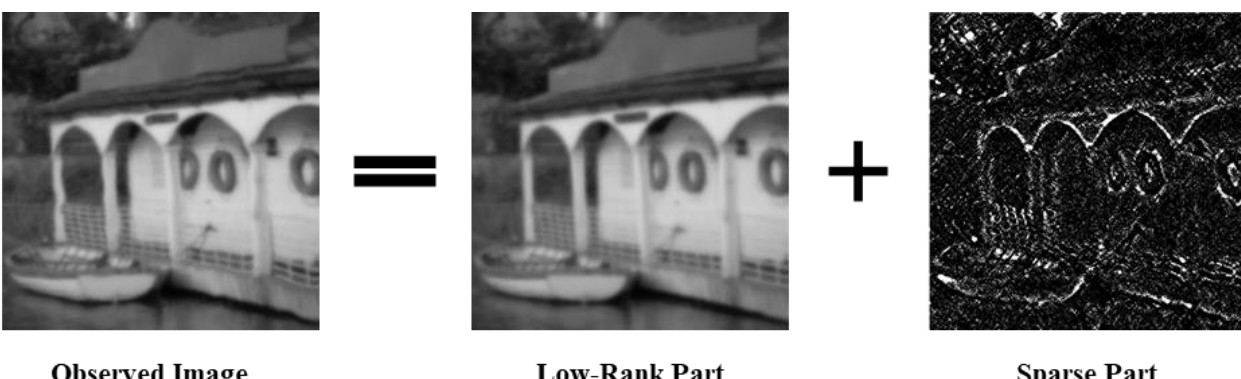

**Figure 2.** Low-rank decomposition of the observed image.

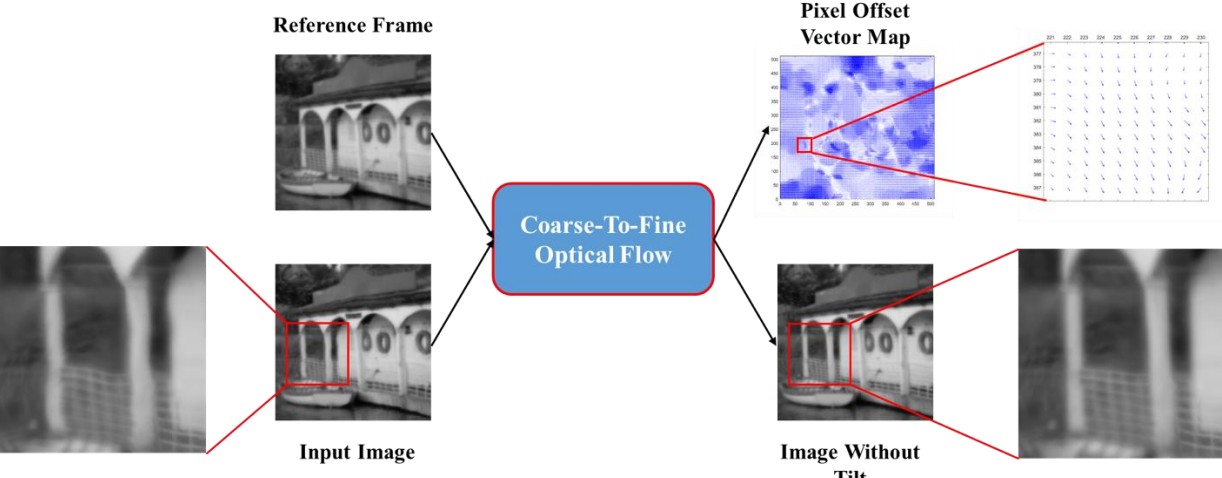

**Figure 3.** Coarse-to-fine optical flow method for image registration.

### 2.3. Data Sets Generation

After obtaining the image without tilt, we must use CNN to remove blur. CNN is a neural network that is specially used to process data with a similar grid structure. Compared with the prior knowledge of the blind deconvolution algorithm, the previous knowledge of CNN was given by training set. Therefore, the data of the training set must conform to the physical rules of atmospheric turbulence. However, in real life, it is challenging to collect images affected and not affected by the atmosphere at the same time. A better solution is to use numerical simulation.

This paper chooses the Zernike polynomial method [24,25] to simulate turbulence-affected images. For an observation system with aperture $D$, the atmospheric phase disturbance φ can be represented by the expansion of the Zernike polynomial:

$$\varphi\left(\frac{D\vec{r}}{2}\right) = \sum_{i=1}^{N} a_i Z_i\left(\vec{r}\right) \tag{7}$$

where $\vec{r}$ is the polar coordinate vector of the unit circle, $a_i$ is the coefficient of the Zernike polynomial, and $Z_i$ is the Zernike polynomial. The Zernike polynomial defined within the unit circle is as follows:

$$\begin{cases} Z_{\text{even}i} = \sqrt{2(n+1)}R_n^m(r)\cos(m\theta)m \neq 0 \\ Z_{\text{odd}i} = \sqrt{2(n+1)}R_n^m(r)\sin(m\theta)m \neq 0 \\ Z_i = \sqrt{2(n+1)}R_n^0(r)m = 0 \end{cases} \tag{8}$$

$$R_n^m(r) = \sum_{s=0}^{(n-m)/2} \frac{(-1)^s(n-s)!}{s![(n+m)/2-s]![(n-m)/2-s]!} r^{n-2s} \tag{9}$$

where $r$ is the polar axis, $\theta$ is the polar angle, $i$ is the sequence number of the Zernike polynomial, $n$ is the radial frequency coefficient, and $m$ is the angular frequency coefficient of the Zernike polynomial. In addition, $n$ and $m$ also satisfy $n \geq m \geq 0$ and $n-m$ being even numbers.

Each term of the Zernike polynomial $Z_i$ is fixed, but the coefficient $a_i$ is variable. We need to determine the relationship between each coefficient that conforms to the laws of atmospheric turbulence. From the energy perspective, Noll [24] provides the covariance between any two Zernike polynomial coefficients $a_i(n_i, m_i)$ and $a_j(n_j, m_j)$:

$$\left\langle a_i, a_j \right\rangle = 2.2802 \left(\frac{D}{r_0}\right)^{5/3} \sqrt{(n_i+1)(n_j+1)} \delta_{m_i m_j} \times \frac{\Gamma\left[\left(n_i+n_j-5/3\right)/2\right] \times (-1)^{(n_i+n_j-m_i-m_j)/2}}{\Gamma\left[\left(n_i-n_j+17/3\right)/2\right]\Gamma\left[\left(-n_i+n_j+17/3\right)/2\right]\Gamma\left[\left(n_i+n_j+23/3\right)/2\right]} \tag{10}$$

$$\delta_{m_i m_j} = \begin{cases} 1, m_i = m_j \\ 0, m_i = m_j \end{cases} \tag{11}$$

where $\Gamma(\cdot)$ is the gamma function, $D$ is the aperture of the observation system, $r_0$ is the atmospheric coherence length, $a_i$ and $a_j$ are the coefficients of the i-th and j-th Zernike polynomials, $m_i$ and $m_j$ are the angular frequency coefficients, $n_i$ and $n_j$ are the radial frequency coefficients, and $\delta_{m_i m_j}$ is the Kronecker function. Using Equation (10), we calculate the covariance matrix of the Zernike polynomial for the first 36 orders, as shown in Figure 4a. Finally, the method of generating point spread function (PSF) conforming to atmospheric turbulence characteristics is shown in Algorithm 1. The process of generating PSF is shown in Figure 4b.

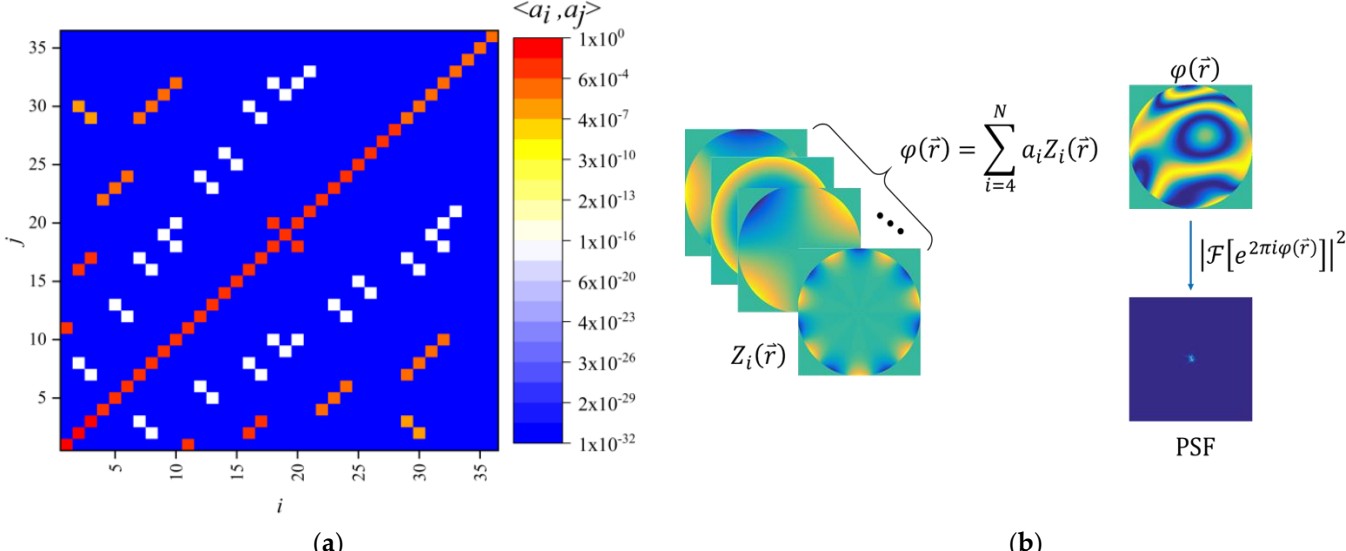

**(a)**            **(b)**

**Figure 4.** Zernike polynomial coefficient covariance matrix and generated PSF diagram. (**a**) Covariance matrix; (**b**) Generated PSF diagram.

| Algorithm 1. Using Zernike polynomials to generate PSF conforming to turbulence characteristics |
| --- |
| **Input:** None |
| **Output:** PSF matrix |
| **STEP 1:** Select the first 36 Zernike polynomials, use Equation (10), and calculate the covariance matrix $C$ of their coefficients |
| **STEP 2:** By using $\sin gular$ value decomposition, obtain $C = VSV^{\mathrm{T}}$, where $S$ is the feature matrix and $V$ is the unitary matrix |
| **STEP 3:** Then simulate and generate random variable $\beta$ satisfying normal distribution, and calculate $\alpha\prime = V\beta$ |
| **STEP 4:** Make $\alpha = (a_1, a_2, \ldots, a_{36}) = \alpha\prime \left(\frac{D}{r_0}\right)^{5/6}$ |
| **STEP 5:** Here we only simulate the blurring caused by turbulence, so we choose the Zernike polynomial coefficients $a_4, a_5, \ldots, a_{36}$ corresponding to $higher-order$ aberrations. Calculate $\varphi$ using Equation (7) |
| **STEP 6:** $\text{PSF} = \left\| \mathrm{F}\left[e^{2\pi i \varphi(\vec{r})}\right] \right\|^2$ |

We noticed that within the range of isoplanatic angle [26], the wavefront distortion caused by turbulence on the atmospheric path is basically consistent. In other words, image blocks within the range of isoplanatic angle can be convoluted using the same PSF. However, the field of view angle of the actual observation system is much larger than the isoplanatic angle. Therefore, different blocks of an image have different corresponding PSFs, as shown in Figure 5.

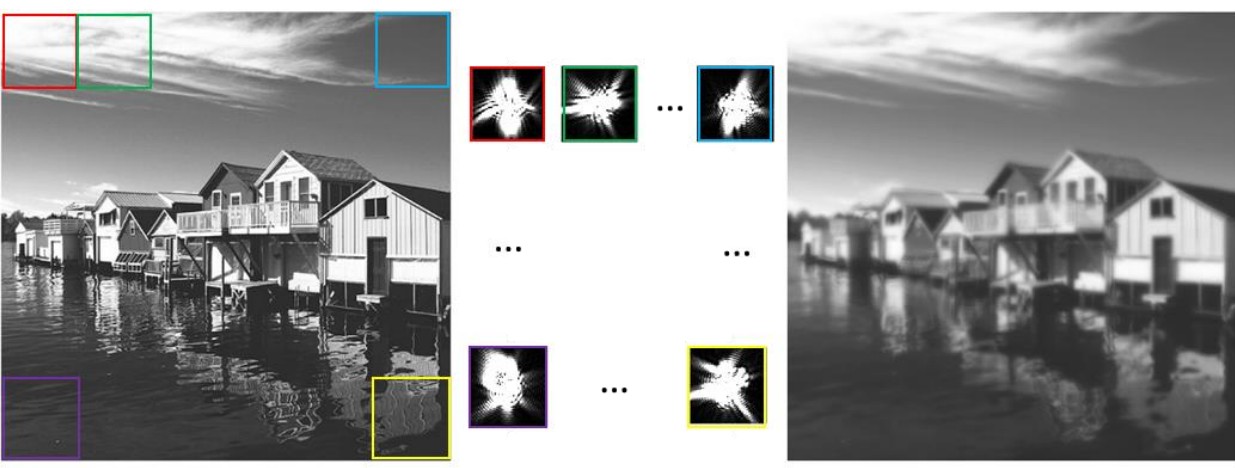

**Figure 5.** Convolutional results of different blocks and PSFs in an image.

Through the above methods, we can simulate and generate a large-scale dataset. The specific parameters of the simulation are given in Table 1. In this paper, the ratio $D/r_0$ of observation aperture $D$ to atmospheric coherence length $r_0$ [27] is used to quantify the effect of turbulence on imaging. Figure 6 shows the simulation results under different conditions.

**Table 1.** Simulator parameters.

| Parameters | Values |
| --- | --- |
| Path length | $L = 7$ km |
| Aperture diameter | $D = 0.305$ m |
| Focal length | $d = 2.438$ m |
| Wavelength | $\lambda = 525$ nm |
| Zernike phase size | $64 \times 64$ pixels |
| Image size | $512 \times 512$ pixels |
| Nyquist spacing (object plane) $\frac{L\lambda}{2D}$ | $\delta_o = 6.0245$ mm |
| Nyquist spacing (focal plane) $\frac{d\lambda}{2D}$ | $\delta_f = 2.0983$ μm |

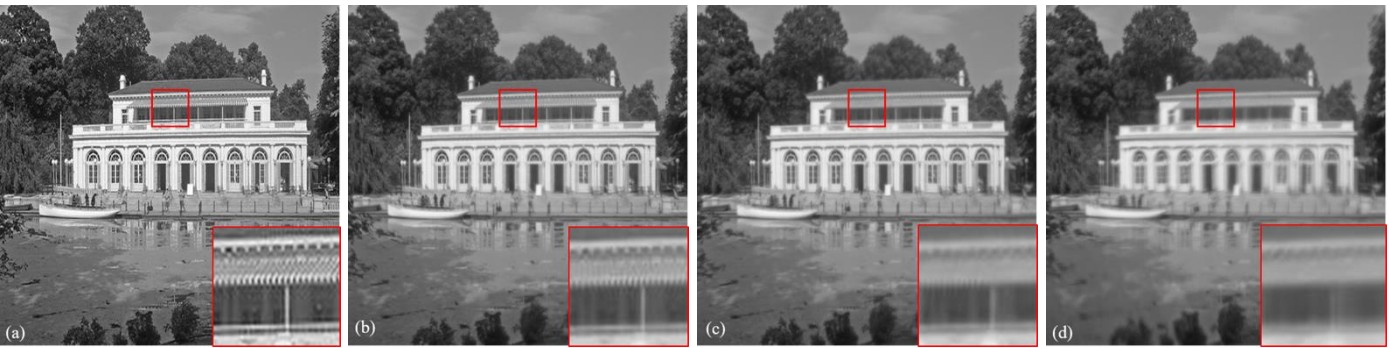

**Figure 6.** Simulation results under different conditions of $D/r_0$: (**a**) clean image; (**b**) $D/r_0 = 2.4312$; (**c**) $D/r_0 = 4.2130$; (**d**) $D/r_0 = 16.7721$.

This paper uses the ImageNet2012 data set to generate 1,491,648 blurred images under $D/r_0 = 2.4312$, $D/r_0 = 4.2130$, and $D/r_0 = 16.7721$ turbulence intensities by the above method. These images are divided into training sets, validation sets, and test sets. The training set contains 1,044,096 images, the validation set contains 149,184 images, and the test set contains 298,368 images. The training sets are used for the training of deep learning network parameters. The validation sets are used for the preliminary evaluation of the model performance during the network training process. The final ability of the network model after training is evaluated by the test sets.

## 2.4. Feature Extraction Technology

For a CNN, its foundation is the feature extraction technology of convolution. The specific implementation method is to perform convolution operations through convolutional (Conv) layer, as shown in Figure 7a. Let $X(i,j)$ be input feature map, $K(i,j)$ be the convolutional kernel. The output feature map $Y(i,j)$ that passes through a convolutional layer can be represented by the following function:

$$Y(i,j) = \sum_m \sum_n X(i-m, j-n)K(m,n) \tag{12}$$

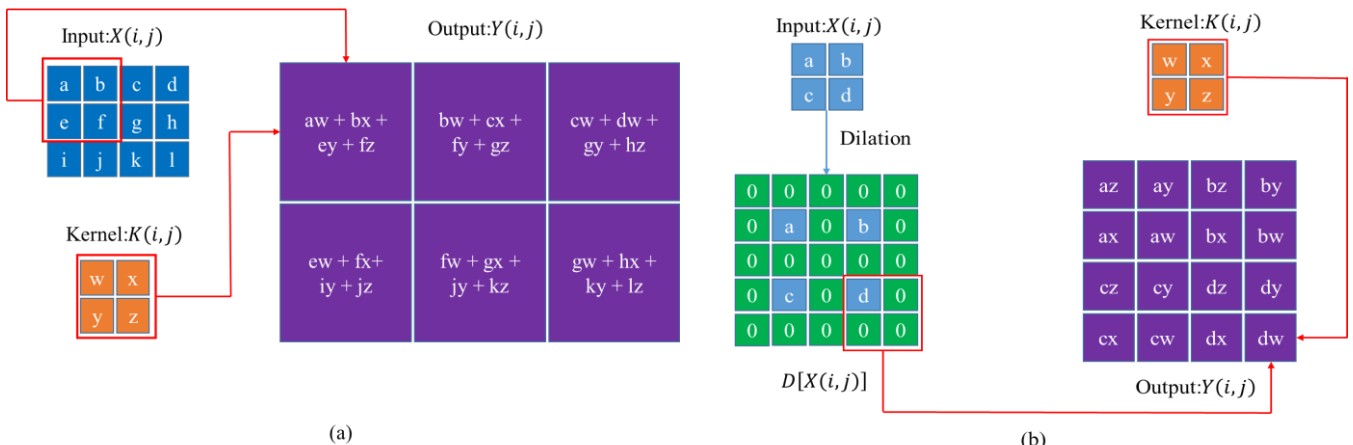

**Figure 7.** The operational logic of Conv and ConvTrans layers in CNNs: (**a**) Conv Layer; (**b**) ConvTrans layer.

After passing through the Conv layer, the feature map must often pass through the activation function Leaky Rectified Linear Unit (LeakReLU) or Rectified Linear Unit (ReLU). CNN needs an activation function to provide nonlinear factors for it so that CNN can approach any nonlinear function. The ReLU and LeakReLU functions are represented as follows:

$$Y(i,j) = \text{ReLU}(X(i,j)) = \begin{cases} X(i,j), X(i,j) > 0 \\ 0, X(i,j) < 0 \end{cases} \tag{13}$$

$$Y(i,j) = \text{LeakReLU}(X(i,j)) = \begin{cases} X(i,j), X(i,j) > 0 \\ -k \cdot X(i,j), X(i,j) < 0 \end{cases}' \tag{14}$$

where $k$ is a constant real number.

The Batch Normalization (BN) layer is also an important layer. Its function is to map the feature map to a distribution with a mean of 0 and a variance of 1. The mathematical expression is as follows:

$$Y(i,j) = \frac{X(i,j) - \text{E}(X(i,j))}{\sqrt{\text{Var}(X(i,j)) + \epsilon}} \times \gamma + \beta \tag{15}$$

where $\text{E}(\cdot)$ and $\text{Var}(\cdot)$ are the mean and variance functions, $\epsilon$ is the variable added to prevent the denominator from appearing as zero. $\gamma$ and $\beta$ are the parameters for affine operation on the input value.

After the above operation, we need to use feature maps for upsampling, usually using transposed convolutional (ConvTrans) layers. The operation of transposed convolution requires dilation of the feature map before convolution, as shown in Figure 7b. Its mathematical expression can be expressed as follows:

$$Y(i,j) = \sum_m \sum_n \text{D}(X(i,j)) K(i-m, j-n) \tag{16}$$

where D is expansion operation.

### 2.5. Structure of Convolution Neural Network

The structure of the deep learning model used in this paper is shown in Figure 8. The model includes 8 Conv layers, 7 LeakyReLU layers, 14 BN layers, 8 ConvTrans layers, 8 ReLU layers, 45 layers in total.

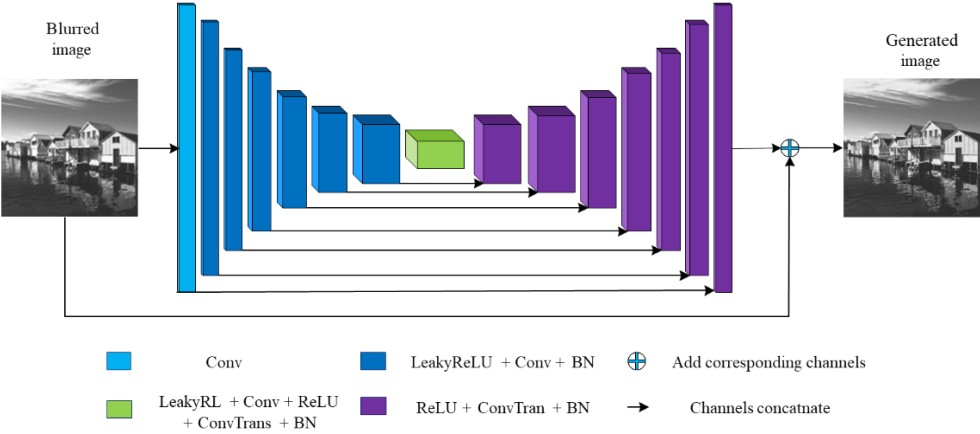

**Figure 8.** CNN structure.

The black arrow in Figure 8 indicates the skip connection operation. Its function is to splice the front and rear feature maps, so as to improve the information acquisition in the up-sampling process and improve the reconstruction of image details by network. The blue plus sign in Figure 8 is the residual structure. Its function is to add the corresponding channels of the two feature maps to get a new image, which is conducive to the training of the depth network.

The input of CNN in this paper is a single-channel two-dimensional image with a size of $512 \times 512$ pixels. After the Conv layer with a convolutional kernel size of $4 \times 4$ and step size of 2, 64 feature maps with a size of $256 \times 256$ pixels are obtained. The second layer is the LeakyReLU layer, with a negative slope 0.2. The number and size of the feature maps have not changed. The third layer is the same as the first layer. After convolution, 128 feature maps with the size of $128 \times 128$ pixels are obtained. The fourth layer is the BN layer, which normalizes the value of each feature map to a mean value of 0 and a variance of 1. After five times of the same LeakyReLU layer, Conv layer and BN layer, 512 feature maps with the size of $4 \times 4$ pixels are obtained. After passing through the LeakyReLU layer and Conv layer of the 20th and 21st layers, the number of feature maps remains unchanged, and the size changes to $2 \times 2$, and the 22nd layer is the ReLU layer. The 23rd layer is the ConvTrans layer. The convolution core size is $4 \times 4$, and the step size is 2. After passing through this layer, 512 feature maps with the size of $4 \times 4$ pixels are obtained. The 24th layer is the same

BN layer. Before entering the 25th layer, the network splices the output feature maps of the 19th layer and the feature maps of the 24th layer to obtain 1024 feature maps with a size of $4 \times 4$ pixels. After seven repetitions of the ReLU layer, CovTrans layer and BN layer, the number of feature maps is compressed to 1, and the size is expanded to $512 \times 512$ pixels. Before the final output, the input image and the network output feature map are added to obtain the generated deblurred image.

### 2.6. Model Hyperparameter Details

The hyper parameters for each layer in our model are shown in Table 2. The input requirement for our network is a $512 \times 512$ grayscale image. The image batch size loaded onto the network is 48. The epochs for network training is 300. The initial learning rate of the network is set to 0.0005. After every 20 epochs of training, the learning rate decreases by 0.5 times. Our network optimizer chooses Adaptive Moment Estimation (Adam). Use the blurred image and the corresponding clear image to calculate the loss function of the network, as follows:

$$\text{Loss}(y, z) = \frac{1}{N} \sum_{i=1}^{N} |y_i - z_i| \tag{17}$$

where $y$ and $z$ are the network-generated and clear images, $N$ is the number of image pixels, $y_i$ and $z_i$ are the pixel values of the network-generated image and the clear image.

**Table 2.** Model hyperparameters.

| Layer Number | 1,3,6,9, 12,15,18,21 | 23,26,29,32, 35,38,41,44 | 2,5,8,11,14,17,20 | 22,25,28,31, 34,37,40,43 | 4,7,10,13,16,19,24,27,30,33,36,39,42,45 |
|---|---|---|---|---|---|
| Type | Conv | ConvTrans | LeakyReLU | ReLU | BN |
| Kernel | $4 \times 4$ | $4 \times 4$ | - | - | - |
| Stride | $2 \times 2$ | $2 \times 2$ | - | - | - |
| Padding | 1 | 1 | | | |
| Negative slope $k$ | - | - | 0.2 | 0 | - |

## 3. Experimental Results and Analysis

### 3.1. Training Results

We use the image training sets of $D/r_0 = 2.4312$, $D/r_0 = 4.2130$, $D/r_0 = 16.7721$ turbulence intensities for the training of CNN. The variation curve of the training loss function is shown in Figure 9. It can be seen from Figure 9 that with the increase of the number of training epochs, the loss functions of the three training sets gradually decrease and finally converge to 0.02065, 0.02403 and 0.03211 respectively. At the same time, the loss functions of the validation sets also gradually decrease and converge to 0.02170, 0.02422 and 0.03221 respectively. Use the test sets to evaluate the restoration effect of the model, as shown in Figure 10.

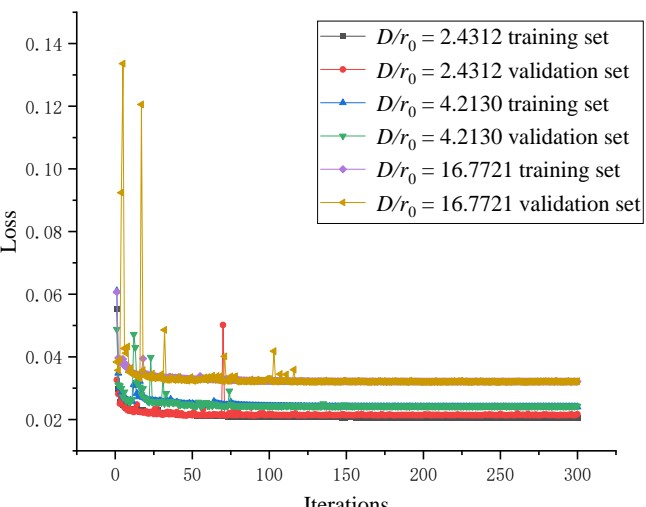

**Figure 9.** Loss function curves of three turbulence intensities.

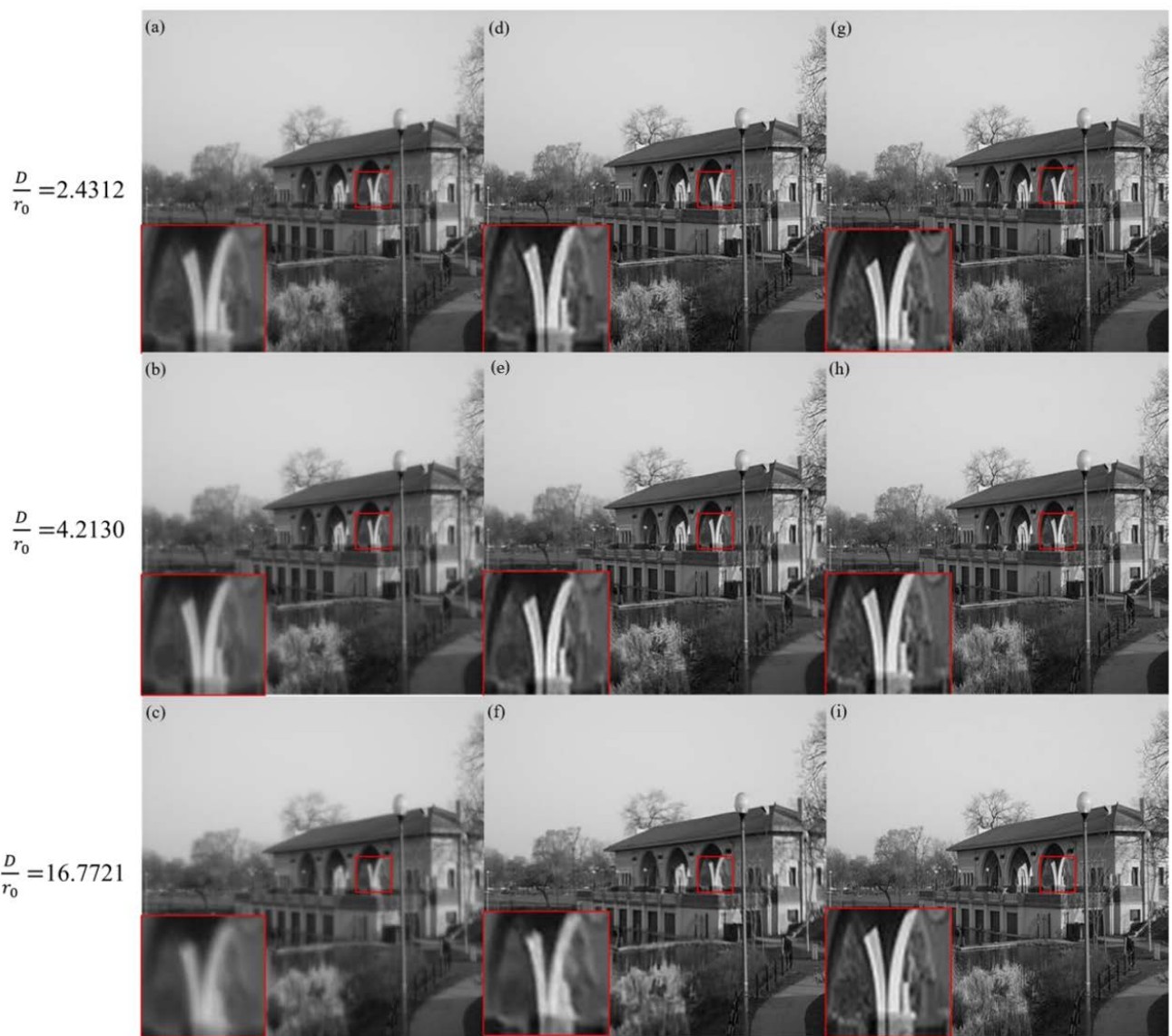

**Figure 10.** Image restoration results under three turbulence intensities: (**a–c**) Turbulence blurred images; (**d–f**) Neural network restored images; (**g–i**) Original clear images.

To verify the effectiveness of the neural network algorithm, the peak signal-to-noise ratio (PSNR) and structural similarity (SSIM) are used as objective evaluation indicators to evaluate the differences between the generated image and the original clear image. The PSNR reflects the difference between the corresponding pixels of the original clear image and the restored image. The higher the PSNR, the sharper the reconstructed image and the better the effect. PSNR is calculated as follows:

$$f_{\text{PSNR}} = 20\lg\left[\frac{\left(2^k - 1\right)}{\sqrt{f_{\text{MSE}}}}\right] \tag{18}$$

where $f_{\text{MSE}}$ is the mean square deviation and $k$ is the number of image bits. SSIM reflects the similarity between the original clear image and the restored image. The closer the value is to 1, the closer the reconstructed image is to the clear image. SSIM calculation equation is as follows:

$$f_{\text{SSIM}}\left(I, I_g\right) = \frac{\left(2\mu_I\mu_{I_g} + c_1\right)\left(2\delta_{I,I_g} + c_2\right)}{\left(\mu_I{}^2 + \mu_{I_g}{}^2 + c_1\right)\left(\delta_I{}^2 + \delta_{I_g}{}^2 + c_2\right)} \tag{19}$$

where $I$ is the original clear image, $I_g$ is the reconstructed image generated by the neural network, $\mu$ is the mean value of the image, $\delta$ is the variance of the image, $\delta_{I,I_g}$ is the covariance of the image, $c_1$ and $c_2$ are constants, set to 6.5 and 58.5. In the experiment, we take the average values of PSNR and SSIM for the test sets, and the results are shown in Table 3. It can be seen from Table 3 that under different turbulence intensities, the PSNR and SSIM of the images reconstructed by neural network are improved compared with the blurred images. For the three turbulence intensities from low to high, the PSNR of the image increased by 6.32 dB, 5.23 dB and 3.11 dB respectively, and the SSIM increased by 22.99%, 20.89% and 15.58% respectively. It can be seen that the reconstructed images have improved in distribution and details compared with the burred images.

**Table 3.** Mean results of PSNR and SSIM under three turbulence intensities.

| Images | Mean PSNR/dB (↑) | Mean SSIM/% (↑) |
|---|---|---|
| Blurred images of $D/r_0 = 2.4312$ | 22.94 | 64.65 |
| Reconstructed images of $D/r_0 = 2.4312$ | 29.26 | 87.64 |
| Blurred images of $D/r_0 = 4.2130$ | 22.91 | 63.78 |
| Reconstructed images of $D/r_0 = 4.2130$ | 28.14 | 84.67 |
| Blurred images of $D/r_0 = 16.7721$ | 22.45 | 59.50 |
| Reconstructed images of $D/r_0 = 16.7721$ | 25.56 | 75.08 |

↑: the PSNR and SSIM of the images reconstructed by neural network are improved compared with the blurred images.

### 3.2. Comparison of Actual Restoration Effects

In the winter of 2022, we chose sunny weather with high visibility for the experiment. We observed ground targets at a distance of 7000 m and obtained 12,521 short-exposure images. The observed environmental parameters are shown in Table 4. After removing the pixel offset in the observed images by optical flow method, we use the trained CNN to restore the registered images. The advanced algorithms currently used for turbulence image restoration include Mao et al. [2] (Optical Flow + Lucky Fusion + Deconv), Chen et al. (CNN) [17], Fazlali et al. (Lucky Fusion + CNN) [18], Hoffmire (Lucky Fusion + CNN) [19], Zhiyuan et al. [20] (CNN). We compared the restored image with the results of the above algorithm, as shown in Figure 11.

**Table 4.** Environmental parameters during observation.

| Parameter | Value |
|---|---|
| Path length | $L = 7$ km |
| Aperture diameter | $D = 0.305$ m |
| Focal length | $d = 2.438$ m |
| Atmospheric coherent length | $r_0 = 0.0229$ m |
| Exposure time | $t = 2$ ms |

It can be seen from Figure 11 that the overall definition and local details of the short-exposure turbulence-degraded image are improved by using our algorithm. Both PSNR and SSIM require the participation of original clear images in the calculation. But in the actual observation process, images that are not affected by the atmosphere cannot be obtained. Therefore, we introduced the Tenengrad function [28]. The Tenengrad function is a commonly used image clarity evaluation function. The larger the value of the Tenengrad function, the higher the clarity of the image. The calculation method for the Tenengard function is as follows:

$$f_{\text{Ten}} = \sqrt{f_x^2 + f_y^2} \tag{20}$$

$$f_x = I_g * \begin{bmatrix} -1 & 0 & 1 \\ -2 & 0 & 2 \\ -1 & 0 & 1 \end{bmatrix}, f_y = I_g * \begin{bmatrix} 1 & 2 & 1 \\ 0 & 0 & 0 \\ -1 & -2 & -1 \end{bmatrix} \tag{21}$$

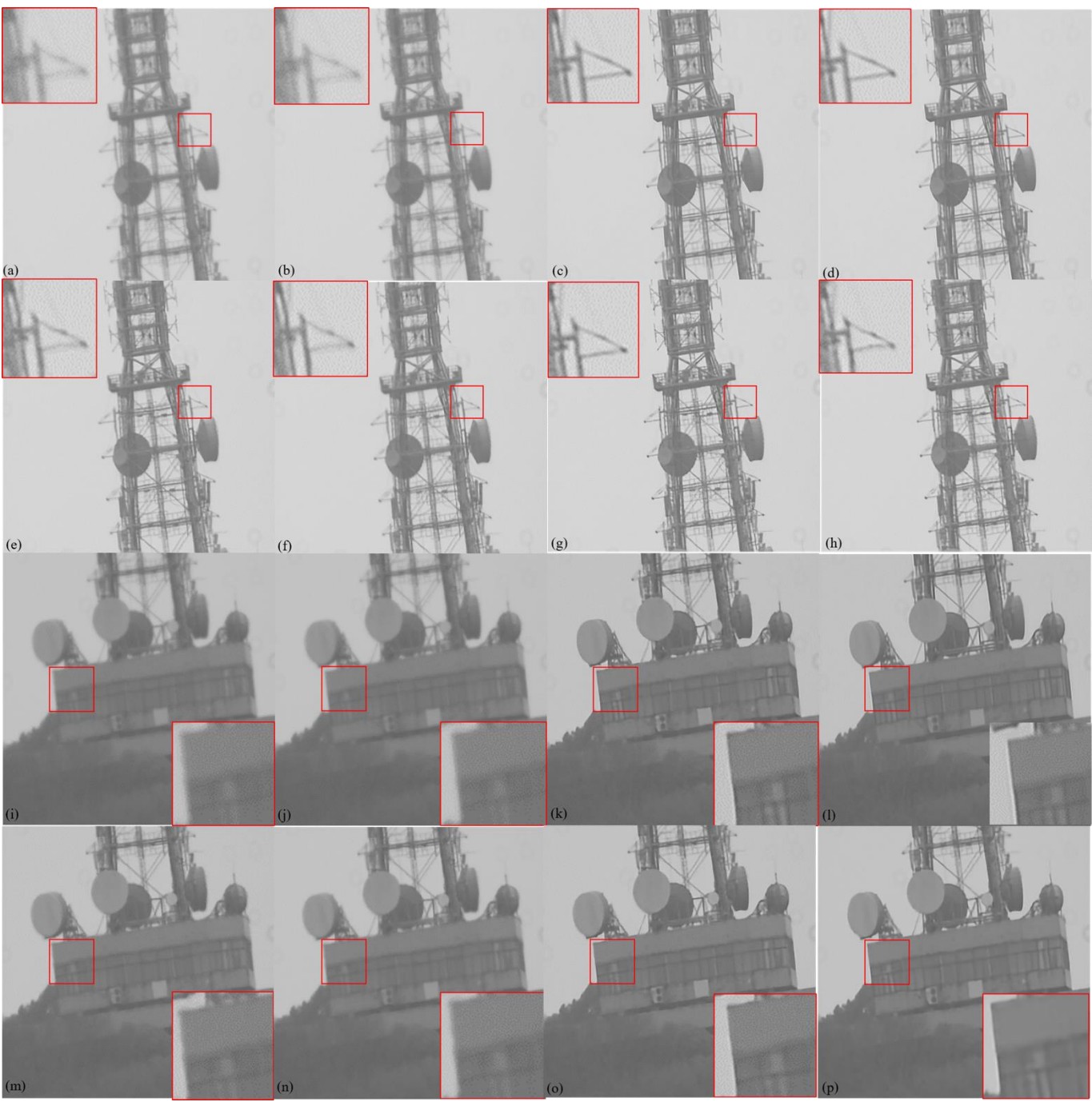

**Figure 11.** Comparison of restoration effects of actual observation images: (**a,i**) original observation images; (**b,j**)Optical flow registration images [4]; (**c,k**) Ours CNN; (**d,l**) Mao et al. [2] (Optical Flow + Lucky Fusion + Deconv);(**e,m**) Chen et al. [17] (CNN);(**f,n**) Fazlali et al. [18] (Lucky Fusion + CNN);(**g,o**) Hoffmire et al. [19] (Lucky Fusion + CNN); (**h,p**)Zhiyuan et al. [20] (CNN).

To demonstrate the correlation between Tenengard and PSNR, we plotted scatter plot of Tenengard and PSNR using simulated restored images, as shown in Figure 12. From Figure 12, there is a significant positive correlation between Tenengard and PSNR. It can also be considered that the larger the Tenengard of the restored image, the larger its PSNR may be, and the closer it is to the original clear image.

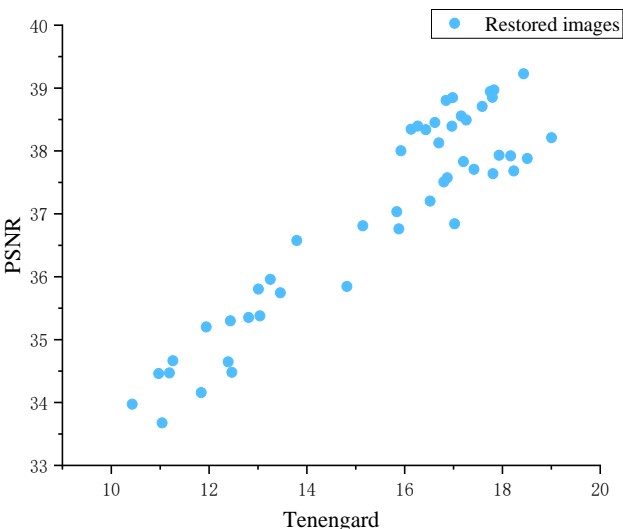

**Figure 12.** Scatter plot of Tenengard and PSNR.

We calculate the Tenengard values for all images in Figure 11, as shown in Figure 13. Overall, our algorithm has slightly higher Tenengard values than other algorithms. In tower body images with a single content, our algorithm significantly improves the image.

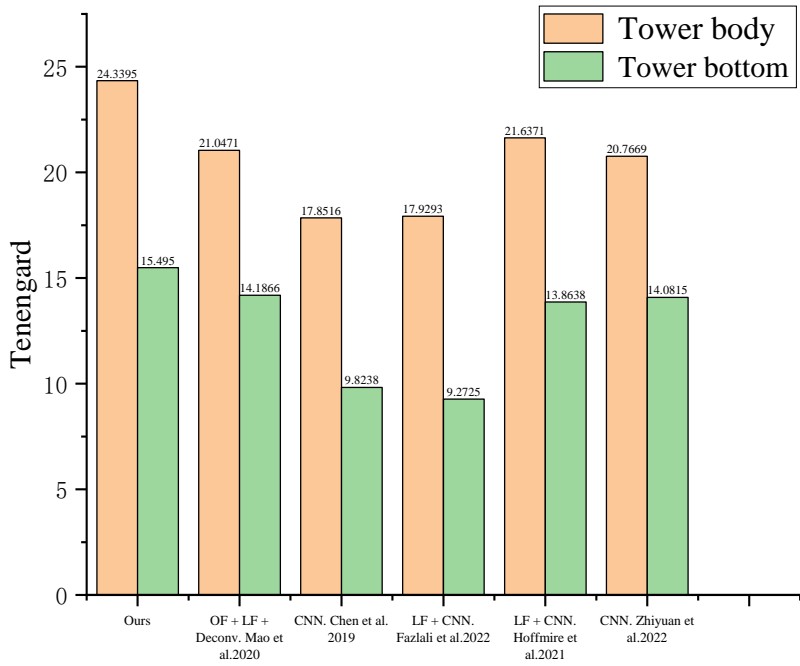

**Figure 13.** Tenengard values of all restored images [2,17–20].

## 4. Discussion

Presently, there are many algorithms for restoring turbulent degraded images, and most of them can achieve good results. However, many conditions limit the application of algorithms. The two main limitations are the demand for computing time and data volume. We have calculated the computational time required abovementioned algorithms to restore 100 images, as shown in Figure 14. From Figure 14, our algorithm has the shortest running time. We replaced the most time-consuming lucky image fusion and blind deconvolution with CNN. At the same time, the CNNs in our algorithm have smaller convolution kernels than other CNNs, and the number of convolutional layers is not particularly large.

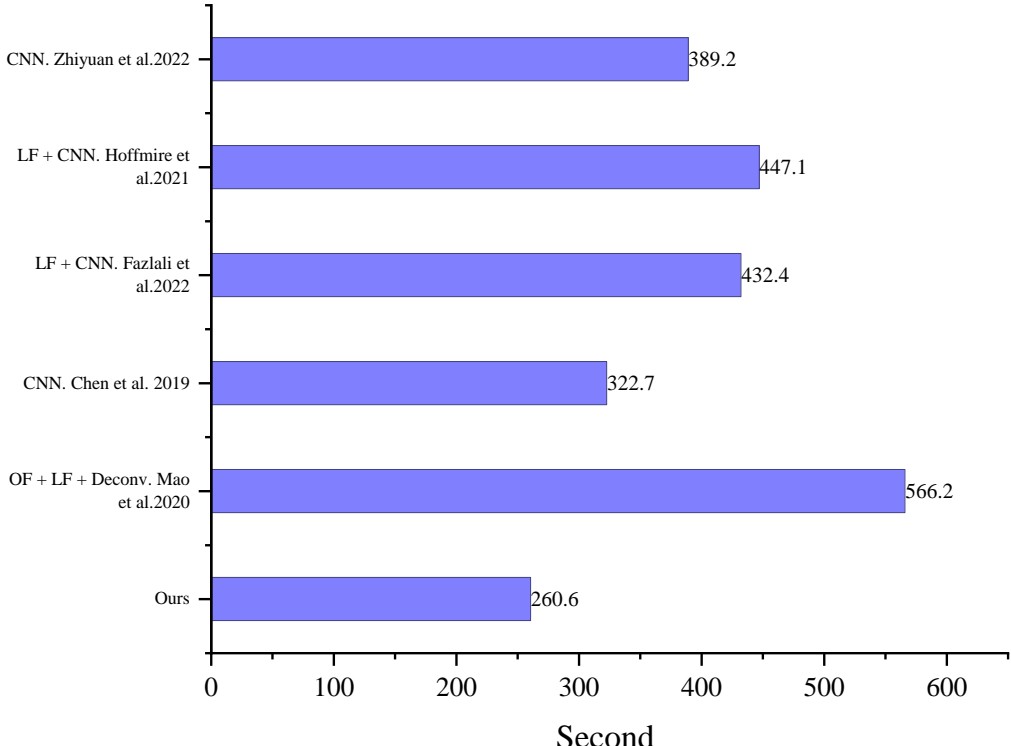

**Figure 14.** Running time for all algorithms to restore 100 images [2,17–20].

Although CNN has certain advantages in computational time, it is not stable enough in image restoration. The basic method for CNN image restoration is to extract image features through different convolutional kernels. In cases where the gradient changes in the image are insignificant, the features extracted by CNN will also be affected. Classifying CNN datasets and training images with less significant gradient changes (also known as more blurry images) separately may be a better method. It can also be understood that reducing the complexity of the dataset can potentially improve the accuracy of deep learning [29,30].

## 5. Conclusions

Image restriction algorithms for the atmosphere are known to be a challenge because it is an ill-posed problem. Traditional lucky image fusion and blind deconvolution require significant computational time. The CNN restoration algorithms have problems such as unstable restoration performance and poor universality. We propose a turbulence mitigation algorithm combining the optical flow method and CNN to solve the above problems. We first use RPCA to extract reference frames from observed images. Then, the optical flow and the reference frame are used to register the restored image. We use the Zernike polynomial method to simulate turbulent degradation datasets. This training set is in line with the turbulence physics model, ensuring that CNN can correctly learn the laws of turbulence. Finally, the trained CNN removes the blurred image after registration.

The experimental results demonstrated that our method could effectively alleviate the degradation of turbulence mitigation. In addition, compared to other methods, our algorithm only requires a shorter computational time to restore high-quality images. With the optimization of algorithm parameters and the update of hardware technology, this algorithm will have wider applications.

**Author Contributions:** Conceptualization, J.C. and W.Z.; methodology, J.C.; software, J.C.; validation, J.C., G.X. and C.Y.; formal analysis, J.C. and C.Y.; investigation, J.C.; resources, G.X. and X.C.; data curation, J.C.; writing—original draft preparation, J.C.; writing—review and editing, W.Z. and J.L.; visualization, J.C.; supervision, W.Z. and J.L. All authors have read and agreed to the published version of the manuscript.

**Funding:** This research received no external funding.

**Institutional Review Board Statement:** Not applicable.

**Informed Consent Statement:** Not applicable.

**Data Availability Statement:** The data presented in this study are available on request from the corresponding author.

**Conflicts of Interest:** The authors declare no conflict of interest.

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
