# Peer review of "Restoration of Atmospheric Turbulence-Degraded Short-Exposure Image Based on Convolution Neural Network"

_photonics, doi:10.3390/photonics10060666_

Round 1

Reviewer 1 Report

It is necessary to incorporate the objective in the abstract, also t is necessary to include the program used for image processing. There is no discussion of results, therefore it is needed to discuss the results with other research works.

Reviewer 2 Report

This study proposes a convolution neural network method for restoring turbulence-degraded short-exposure images. The model uses a U-net network and residual structures and is trained using a massive dataset simulated by the Zernike polynomial. The method has good generalization and outperforms current blind de-convolution and convolution neural network algorithms. However, before further consideration of the manuscript, the authors must “fully” address the comments listed below:

1-    The abstract requires restructuring for improved clarity. A logical order should begin with an overview of the topic and rationale, followed by the methodology used and manuscript outline. The results need to be detailed, followed by implications and future directions. 

2-    Can the proposed method be applied to long-exposure images, and how does its performance compare to other methods? 

3-    Can the proposed method be further optimized for real-time applications, and what is its expected computational complexity? What are the main limitations of existing methods for restoring turbulence-degraded images, and how does the proposed method address these limitations? 

4-    The text mentions that the method uses a massive dataset simulated by Zernike polynomial for supervised learning, but it does not provide details on how the hyperparameters of the model were optimized.

5-    The authors mentioned that “Compared with the current advanced blind de-convolution algorithm, the demand of this algorithm for the measured data is reduced from hundreds of images to a single image, and the operation time is 19 reduced from 566s to 2s”. However, this statement is partially true because the dataset might be highly complex such that even a better network/algorithm may not necessarily improve the model accuracy. In machine learning, this can refer to “Kolmogorov complexity” denoting the length of the shortest computer program that produces the output. Therefore, write a paragraph in your paper arguing that reducing the complexity of your dataset can potentially improve the accuracy of the deep learning model and reference the 2 papers listed below (that reduce the complexity of their dataset to improve the accuracy of their machine learning models)

·      Bolon-Canedo, V., & Remeseiro, B. (2020). Feature selection in image analysis: a survey. Artificial Intelligence Review, 53(4), 2905-2931.

·      Kabir, H., & Garg, N. (2023). Machine learning enabled orthogonal camera goniometry for accurate and robust contact angle measurements. Scientific Reports, 13(1), 1497.

Reviewer 3 Report

[1]- The blind deconvolution algorithm requires hundreds of images to participate in the operation, and the operation time is long, and the average time is up to 566 seconds. The proposed algorithm only needs a single image to complete the restoration, and the average calculation time is 2 seconds, which is more advantageous in the amount of data required and the operation time. The validation and realization to verify the effectiveness of the proposed scheme need a better and more detailed explanation. It can be seen from Figure 6 that the overall definition and local details of the short-exposure turbulence-degraded image are improved by using CNN. Please the authors clarify the significance of this study. This part requires substantial improvement. And show the comparison with other machine learning approaches and deep Learning.

This paper can be accepted if the following comments are addressed. The following comments are for further consideration. Comments and Suggestions for Authors:

This paper proposes a method based on a convolution neural network for the restoration of turbulence-degraded short-exposure images. This model is a kind of autoencoder neural network model, which is composed of a U-net network and residual structures.

[2]- Some simulation or experiment results of comparison should be added. Some improvement in English usage is needed. The conclusion must be rewritten as it does not match the abstract.  It must reflect all the graphs. The introduction section is informative and clearly states the problem that the study aims to solve. However, some sentences are written in a passive voice and could be rephrased to make them more active and engaging.

[3]- The method section is well-structured and provides sufficient information on how you conducted the study. However, some technical terms are not defined and could be explained briefly for clarity.

[4]- The results and discussion section are informative and well-organized. However, the section could be improved by including a more detailed analysis and comparison of your method's performance with that of other methods. Additionally, some of the figures are not clear, and better-quality figures could improve readability.

[5]- The readability and English grammar should be improved

[6]- Proofreading and editing could improve the manuscript's readability.

[7]- The methodology section could benefit from the further explanation of some of the techniques used. For example, what is the "Zernike polynomial method," and how was it used to generate the training sets?

[8]- More information could be given about the limitations of the study, especially in the discussion section. What are the potential sources of error, and how might they affect the results?

[9]- The manuscript would benefit from more detailed and clear descriptions of the statistical methods used to analyze the data.

[10]- Clarification and justification are required for the effectiveness of the neural network algorithm, the peak signal-to-noise ratio (PSNR) and structural similarity (SSIM) are used as objective evaluation indicators to evaluate the quality of the generated image.

[11]- The figure legend quality is very poor. Authors should increase the figure legend quality.

[12]- In the experiment, the author takes the average values of PSNR and SSIM for the test sets, and the results are shown in Table 2. It can be seen from Table 2 that under different turbulence intensities,  the PSNR and SSIM of the images reconstructed by the neural network are improved compared with the blurred images. the different turbulence intensities for weak, moderate, and strong turbulence and can investigate all turbulences interstates and show the turbulence strength for refractive index structures and compare with all possible solutions to improve performance. Comparisons should be improved. It needs some modifications and improvements. Investigate all possible solutions and achieves a smaller loss value and converge faster.

Reviewer 4 Report

The method proposed in this paper is to train the parameters of the CNN through a large number of training sets, so as to achieve the effect of restoration of turbulence-degraded short-exposure images. This method is used to replace the traditional lucky image fusion and blind deconvolution. Compared with the traditional method, it has less dependence on the amount of measured data and greatly reduces the calculation time. In the measured data, compared with the current convolution neural network algorithms, its restoration performance is better. The paper is interesting but the presented description of research activities, that were provided by the authors, are too limited. The authors write about Zernike-based polynomial method but the practically don't pay attention to this method in the paper! How was it applied? When was it applied? What is the relation between Figure 2 and mentioned Zernike-based method? All these points should be clearly presented. Moreover, Zernike-based CNN approaches are well-known and very often presented in the papers. For example:

https://www.mdpi.com/2227-7390/9/20/2616

Then, what is the scientific novelty of the paper?

The paper lacks mathematical backgroud.

Where is the full pipeline of the proposed solution? Do the authors use any image preprocessing methods? 

The authors present in Table 2 image metrics for image quality estimation. It is very good but it is not enough. The authors should present, for example, a dependence between a level of PSNR and a quality of the proposed solution in terms of metrics presented Table 4 and in terms of SSIM of the final images.

Figure 3 is too trivial.

Table 4 should be better commented on. How were presented metrics calculated? Did the authors use only CNN-based methods for the comparison?

The use of terms 'learning' and 'training' is confused.

The paper should be proofread. 

Round 2

Reviewer 2 Report

The authors addressed my comments and the manuscript can be published in the current format. 

Author Response

Thank you for your affirmation and support of this paper.

Reviewer 4 Report

The authors addressed practically all my concerns.  

The point with the PSNR and its relation with the final quality of the proposed solution is still not presented. Please, address this point.
